# Beyond its preferential niche: *Brucella abortus* RNA down-modulates the IFN-γ-induced MHC-I expression in epithelial and endothelial cells

**Agustina Serafino[1], Yasmín A. Bertinat[1], Jorgelina Bueno[1], José R. Pittaluga[1], Federico Birnberg Weiss[1], M. Ayelén Milillo[2,3]☯*, Paula Barrionuevo[1]☯**

**1** Instituto de Medicina Experimental-Consejo Nacional de Investigaciones Científicas y Técnicas, Academia Nacional de Medicina; Buenos Aires, Argentina, **2** Universidad Nacional de Río Negro. Instituto de Estudios en Ciencia, Tecnología, Cultura y Desarrollo. Río Negro, Argentina, **3** Consejo Nacional de Investigaciones Científicas y Técnicas. Argentina

☯ These authors contributed equally to this work.
* mililloayelen@gmail.com

## Abstract

*Brucella abortus* (*Ba*) is a pathogen that survives inside macrophages. Despite being its preferential niche, *Ba* infects other cells, as shown by the multiple signs and symptoms humans present. This pathogen can evade our immune system. *Ba* displays a mechanism of down-modulating MHC-I on monocytes/macrophages in the presence of IFN-γ (when Th1 response is triggered) without altering the total expression of MHC-I. The retained MHC-I proteins are located within the Golgi Apparatus (GA). The RNA of *Ba* is one of the PAMPs that trigger this phenomenon. However, we acknowledged whether this event could be triggered in other cells relevant during *Ba* infection. Here, we demonstrate that *Ba* RNA reduced the surface expression of MHC-I induced by IFN-γ in the human bronchial epithelium (Calu-6), the human alveolar epithelium (A-549) and the endothelial microvasculature (HMEC) cell lines. In Calu-6 and HMEC cells, *Ba* RNA induces the retention of MHC-I in the GA. This phenomenon was not observed in A-549 cells. We then evaluated the effect of *Ba* RNA on the secretion of IL-8, IL-6 and MCP-1, key cytokines in *Ba* infection. Contrary to our expectations, HMEC, Calu-6 and A-549 cells treated with *Ba* RNA had higher IL-8 and IL-6 levels compared to untreated cells. In addition, we showed that *Ba* RNA down-modulates the MHC-I surface expression induced by IFN-γ on human monocytes/macrophages via the pathway of the Epidermal Growth Factor Receptor (EGFR). So, cells were stimulated with an EGFR ligand-blocking antibody (Cetuximab) and *Ba* RNA. Neutralization of the EGFR to some extent reversed the down-modulation of MHC-I mediated by *Ba* RNA in HMEC and A-549 cells. In conclusion, this is the first study exploring a central immune evasion strategy, such as the downregulation of MHC-I surface expression, beyond monocytes and could shed light on how it persists effectively within the host, enduring unseen and escaping CD8[+] T cell surveillance.

**Data Availability Statement:** All relevant files are available from the Reppositorio Institucional

   

CONICET digital database (http://hdl.handle.net/11336/237492).

**Funding:** This study was financially supported by the Fondo para la Investigación Científica y Tecnológica (FONCYT) of the Agencia Nacional de Promoción de la Investigación, el Desarrollo Tecnológico y la Innovación, Argentina, in the form of grants awarded to MAM (PICT-2020 SERIE A-00882) and PB (PICT-2020 SERIE A-00978). No additional external funding was received for this study. The funder had no role in study design, data collection and analysis, decision to publish, or preparation of the manuscript.

**Competing interests:** The authors have declared that no competing interests exist.

## Introduction

*Ba* is one of the intracellular bacterial species responsible for brucellosis, a chronic infectious disease handling a significant global impact [1, 2]. This condition is multisystemic affecting different tissues and organs [1]. Brucellosis has been classified by the World Health Organization as one of the seven "neglected endemic zoonotic diseases of particular interest" [3, 4]. In fact, there is no licensed vaccine for use in humans. Furthermore, current livestock vaccines using live attenuated strains are known to cause human illness and occasionally abortion in vaccinated livestock and are therefore not a suitable solution for eradicating the disease on a large scale [5, 6]. Therefore, the deep understanding of the mechanisms and strategies employed by *Ba* to persist in the host is -and will be- crucial to the design of effective vaccines and treatments to control the disease in humans and other animals.

The infection in humans triggers the innate and adaptive immunity towards a Th1 profile with the consequent activation of CD8$^+$ T cells [7–9]. Despite this response, this bacterium employs several immune evasion strategies to chronically persist in host cells [10]. Previous results from our laboratory demonstrate that one of the mechanisms displayed by *Ba* is the decrease of MHC-I on monocytes/macrophages surface when Th1 response is being held, *i.e.*, in the presence of IFN-γ [11, 12]. As a result, the infected cells have a decreased capacity for antigenic presentation to CD8$^+$ T cells [11, 12]. Moreover, we elucidated that *Ba* RNA, a pathogen-associated molecular pattern (PAMP) associated with bacterial viability (*vita*-PAMP), is a component involved in the diminished MHC-I expression [13, 14]. In addition, human TLR8, an endosomal receptor capable of sensing single-stranded RNA and RNA degradation products, is the receptor involved in MHC-I decrease [13, 14]. Regarding the signaling pathway, we demonstrated that *Ba* RNA-mediated down-modulation of MHC-I expression is dependent on the EGFR pathway [13–15]. The decrease of MHC-I surface expression in monocytes/macrophages is not due to transcriptional and/or translational inhibition, instead these proteins are retained within the Golgi Apparatus (GA) [11].

As mentioned above, we have made substantial advances in the understanding of a central strategy for the bacterium: the downregulation of MHC-I expression. However, these studies only focused on the preferential intracellular niche for *Ba*, the monocyte/macrophage. As it is already known, *Ba* can also infect and replicate in different cell types such as lung epithelial and endothelial cells, osteoblasts, synoviocytes, hepatocytes, astrocytes, and microglia, among others [16–24]. Considering that *Ba* may be acquired by humans through the inhalation of contaminated aerosols, the lung epithelium plays an important role in brucellosis. This epithelium is the first barrier that *Brucella* must overcome when infecting the respiratory system. *Ba* can trigger an inflammatory response in said pulmonary epithelium and generate different clinical manifestations, including empyema, pleural effusion, granulomas and solitary nodules, interstitial pneumonia, hilar and paratracheal lymphadenopathy, and pneumothorax. This is why the study of lung epithelium must be considered in the diagnosis as well as the management of the disease [25, 26]. Ferrero *et al.* demonstrated that lung epithelium plays an active role in the immune response against brucellosis secreting inflammatory mediators such as IL-8 y MCP-1, which contribute to the host immune response [16, 17].

On the other hand, within the 5% mortality rate of human brucellosis, complications from endocarditis account for the largest share. Moreover, *Brucella* has been frequently isolated from affected aortic and/or mitral valves [2, 27, 28]. This is why studying the role of the endothelial cells during *Ba* infection is key to understand the pathogenesis and the progression to a chronic condition. Ferrero *et al.* [18] has showed that *Brucella* spp. infect and survive in endothelial cells and can induce a proinflammatory response which may be involved in the vascular manifestations of patients with brucellosis.

Despite the crucial relevance of lung epithelial and endothelial cells in the context of brucellosis pathogenesis, up to now, there has not been studies of *Ba* immune evasion strategies on those cells. This research aims to investigate the long-term persistence of the disease in various organs of the infected host. It is the first study of its kind.

Here, we study for the first time whether *Ba* triggers strategies of immune evasion in other cell types and the effect of a drug already used in other diseases which could revert MHC-I down-modulation. More specifically, we assessed the effect of the RNA of *Ba* on the expression of MHC-I, as well as the preliminary mechanism and pathway involved in lung epithelial and endothelial cells. As model target cells we used A-549 (a human alveolar epithelium cell line), Calu-6 (a human bronchial epithelium cell line), and HMEC (an endothelial microvasculature cell line). We first determined whether *Ba* RNA was able to decrease MHC-I surface expression in all cell lines evaluated. We also investigated whether this phenomenon is caused by the retention of MHC-I in the GA, as it happens for human macrophages. Finally, we explored the capacity of Cetuximab, an inhibitor of the EGFR pathway, to revert the decrease of MHC-I surface expression. We present the results of this first study.

## Materials and methods

### Bacteria strains

*Brucella abortus* S2308 was cultured in triptein soy agar medium. The number of bacteria on stationary-phase cultures was determined by comparing the optical density (OD) at 600 nm with a standard curve. All live *Brucella* manipulations were performed in Biosafety Level 3 laboratory (BSL3) of the Operational Unit Center for Biological Containment (UOCCB) of the National Administration of Laboratories and Health Institutes 'Dr. Carlos G. Malbrán' (ANLIS-Malbrán), and of the Instituto de Investigaciones Biomédicas en Retrovirus y SIDA (INBIRS, UBA-CONICET) thanks to an agreement established in 2017 and 2002, respectively.

### Cell cultures

For all the experiments, THP-1, Calu-6, A-549 and HMEC cell lines were used. All experiments were performed at 37˚C in a 5% $CO_2$ atmosphere and a standard medium consisting in RPMI 1640 for THP-1 cells and HMEC or DMEM (Gibco) for A-549 and Calu-6 cells, supplemented with 25 mM Hepes, 2 mM L-glutamine, 10% heat-inactivated FBS (Thermo Fisher Scientific), 100 U penicillin/ml, and 100 μg streptomycin/ml. THP-1, Calu-6, A-549 and HMEC cells were obtained from the American Type Culture Collection (Manassas, VA, USA) and cultured, as previously described [29–31]. THP-1 cells were treated with 1 ng/ml PMA (Sigma-Aldrich) for 24 h to promote adherence. Then they were washed, and complete medium was added for 72 h. For all the experiments carried out, the cell confluence was 80%. This study was approved by Academia Nacional de Medicina's Ethics Committee (number: 36/23/CEIANM).

### RNA preparation

$5 \times 10^9$ CFU of *Ba* were resuspended in 1 ml of TRIzol (Invitrogen) and RNA was purified using Quick-RNA DirectZol columns (Zymo Research) according to the manufacturer's instructions. The quantification and purity of the RNA was determined using a DeNovix DS-11 spectrophotometer (DeNovix Inc.). In all cases, the absorbance ratio 260/280 was higher than 2.0, and the ratio 260/230 was higher than 1.8.

## Viability assay

For viability assays, THP-1, Calu-6, A-549 and HMEC were treated with *Ba* RNA in the presence of IFN-γ for 48 h. Cells were washed with PBS buffer and resuspended in diluted 100 μl Zombie Violet™ solution to check the viability of the treated cells. Heat-killed cells were included as a positive control of the technique. Then, cells were incubated at room temperature, in the dark, for 15–30 minutes. The cells were then washed and surface antibody staining procedure was performed. After labelling, cells were analyzed with FACSCalibur (BD Biosciences) or Partec CyFlow (LabSystems) flow cytometers. Data was analyzed with FlowJo V10 software (FlowJo, LLC).

## *In vitro* stimulation

$2.5 \times 10^5$ THP-1 cells were treated with PMA (1 ng/ml) for 24 h to promote adherence. Then cells were washed, and complete medium was added for 72 h. Calu-6, A-549 and HMEC were incubated in 48-well plates ($6 \times 10^4$ cells/well) for 24 h to promote adherence. Then, cells were stimulated with *Ba* RNA in the presence of IFN-γ for 48 h at 37˚C in a 5% $CO_2$ atmosphere. In all cases, the surface expression of MHC-I or the intracellular expression of TLR8 was evaluated by flow cytometry, as described in the following paragraph.

## Flow cytometry

THP-1, Calu-6, A-549 and HMEC cells were stained with an anti-human MHC-I conjugated to FITC (W6/32) antibody or the corresponding isotype control antibody. All cell lines were permeabilized with Triton (0.1%) before being stained with an anti-human TLR8 conjugated to PE (clone 44C143; Invitrogen) antibody or the corresponding isotype control antibody. After labelling, cells were analyzed on a FACSCalibur flow cytometer (BD Biosciences) or Partec CyFlow (LabSystems) and data were processed with the FlowJo V10 applications (FlowJo, LLC). Data was normalized to untreated cells, *i.e.*, in each set of experiments, the MFI of treatments (from the % of positive cells for the marker) was divided by MFI of media condition (from the % of positive cells for each marker).

## Confocal microscopy

$2.5 \times 10^5$ THP-1 cells were treated with 1 ng/ml PMA for 24 h to promote adherence in chamber-slides. Then they were washed, and complete medium was added for 72 h. $6 \times 10^4$ cells/well Calu-6, HMEC and A549 cells were incubated in chamber-slides for 24 h to promote adherence. Then, cells were stimulated with *Ba* RNA in the presence of IFN-γ for 48 h, fixed with 2% paraformaldehyde, permeabilized with 0.1% saponin and incubated with anti-HLA-ABC class I mAb W6/32 (BioLegend), and an Alexa 546-labelled secondary Ab (Invitrogen). Golgi apparatus was detected using a mAb specific for GM130 (BD Biosciences) following Alexa 488-labelled secondary Ab (Invitrogen). For nuclear staining, TOPRO-3 (Thermo Fisher Scientific) was used. Slides were mounted with PolyMount (Polysciences) and analyzed using FV-1000 confocal microscope with an oil-immersion Plan Apochromatic 60X NA1.42 objective (Olympus).

## Determination of cytokine concentration

The concentration of different cytokines and chemokines (IL-8, IL-6 and MCP-1) was quantified in supernatants (SNs) from all cell lines stimulated with *Ba* RNA in the presence of IFN-γ for 48 h by ELISA sandwich, according to the manufacturer's instructions. The following detection kits were used: IL-6 (e-Bioscience), MCP-1 (BioLegend) and IL-8 (BioLegend).

### Reagents

Antibody targeting EGFR (Cetuximab) was purchased from Merck Serono. The final concentration used was 50 μg/ml in all experiments.

### Statistical analysis

Results were analyzed with one-way ANOVA followed by *post hoc* Tukey test or two-way ANOVA followed by *post hoc* Bonferroni test with GraphPad Prism software, 8.4.3 version.

## Results

### *Ba* RNA diminishes MHC-I expression on different cell lines

To evaluate whether MHC-I down-modulation mediated by *Ba* RNA were mimicked in other cells able to be infected by *Ba*, cells from the human bronchial epithelium cell line (Calu-6), cells from the human alveolar epithelium cell line (A-549) and cells from the endothelial microvasculature cell line (HMEC) were stimulated with *Ba* RNA at different doses in the presence of IFN-γ for 48 h. Afterwards, MHC-I surface expression was determined by flow cytometry. As an experimental control, THP-1 cells were used. As shown, *Ba* RNA diminished MHC-I surface expression in all four cell lines (Fig 1A–1D). Cell viability was assessed using a

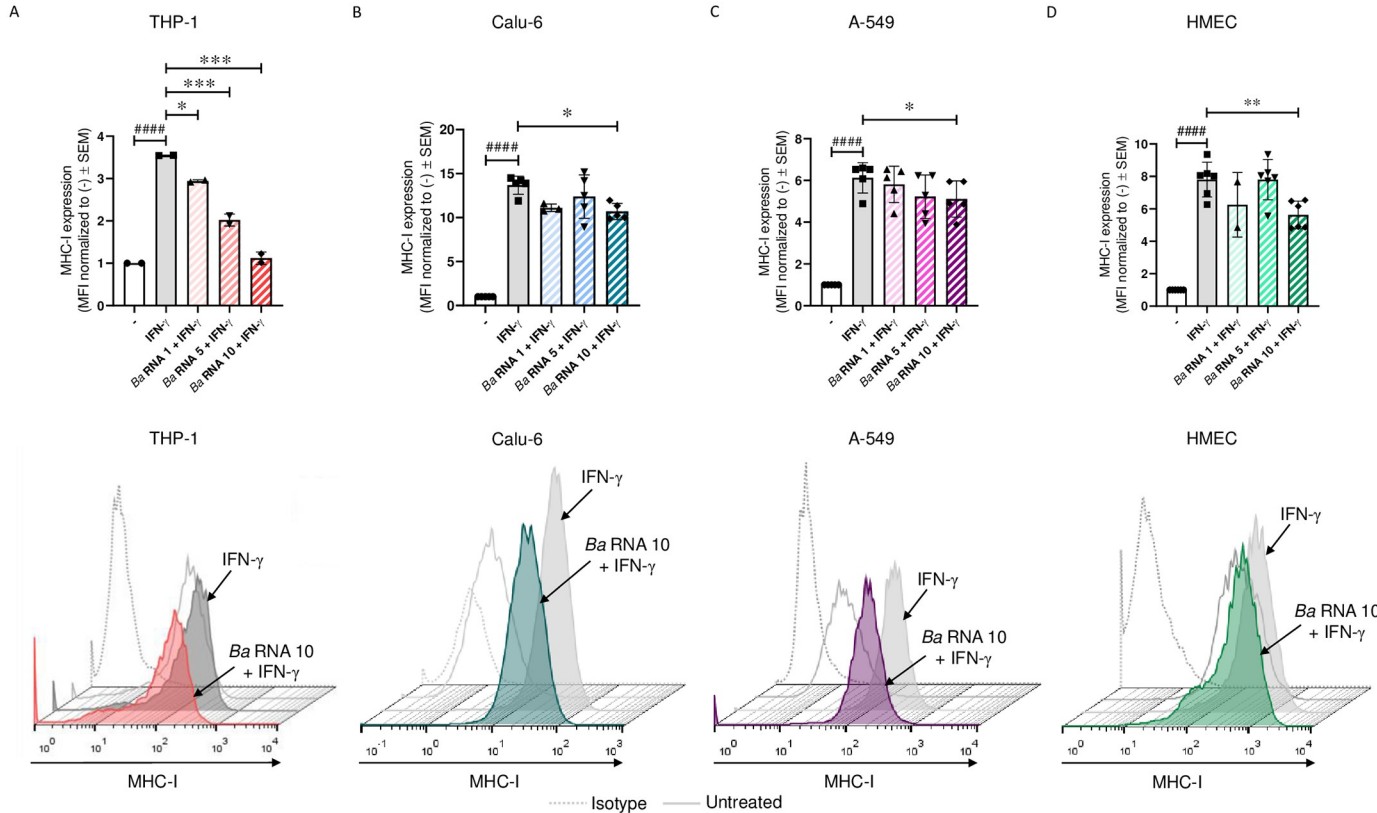

**Fig 1. *Ba* RNA diminishes the IFN-γ induced MHC-I expression on Calu-6, A-549 and HMEC.** (A) THP-1, (B) Calu-6, (C) A-549 and (D) HMEC were attached on a 48-well plate for 24 h. Then, the cells were stimulated with three doses of *Ba* RNA (1, 5 and 10 μg/ml) in the presence of IFN-γ. After 48 h, flow cytometry was used to detect MHC-I expression. IFN-γ-treated cells were used as a positive control. Bars represent the geometric means normalized to untreated cells ± SEM of at least two independent experiments. MFI, mean fluorescence intensity. -, untreated cells. $^{####}P<0.0001$ *vs* untreated cells. $^{*}P<0.05$; $^{**}P<0.01$; $^{***}P<0.001$ *vs* IFN-γ-treated cells.

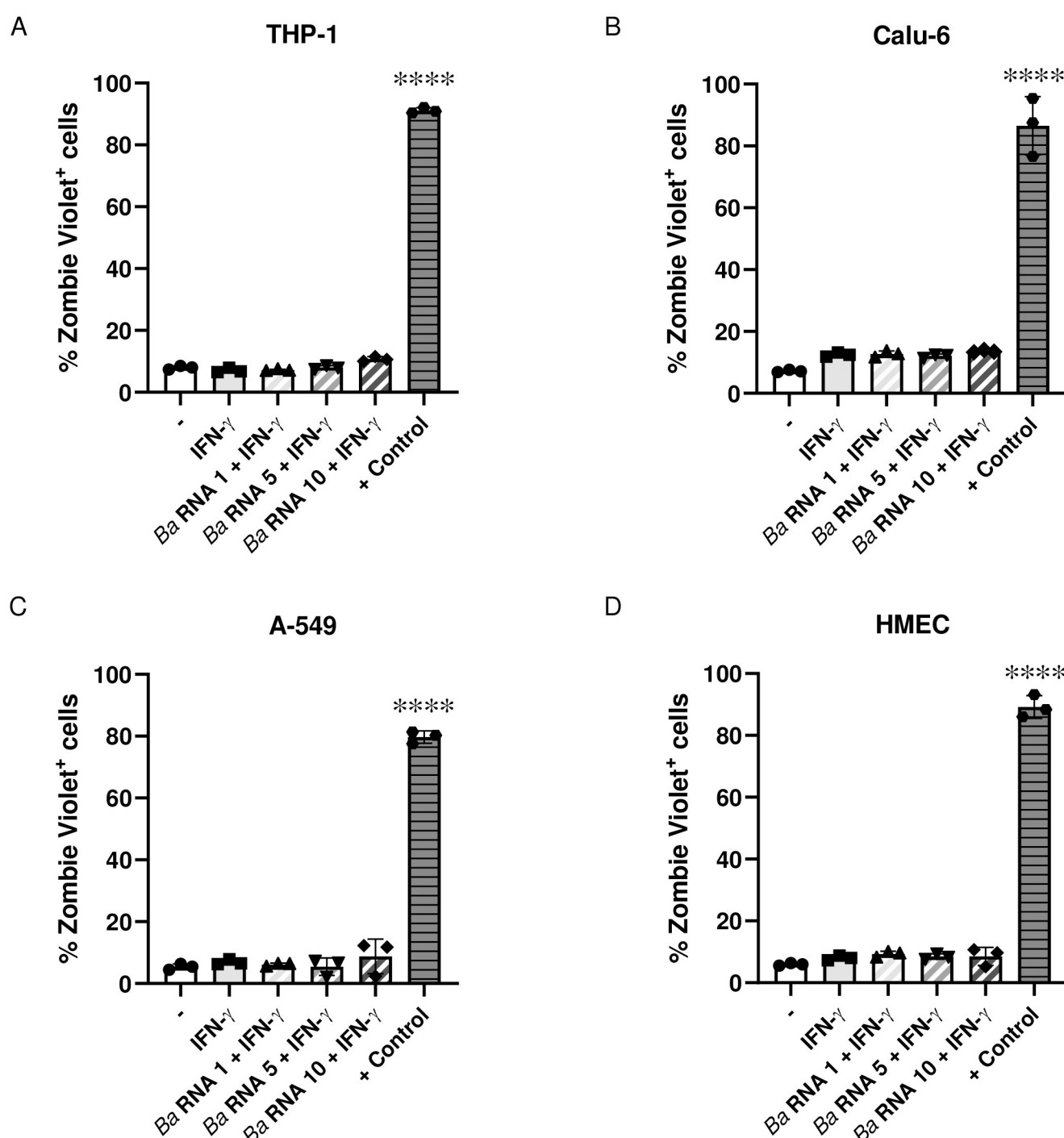

**Fig 2. The MHC-I decrease mediated by *Ba* RNA is not due to a loss of cell viability.** (A) THP-1, (B) Calu-6, (C) A-549 and (D) HMEC were attached on a 48-well plate for 24 h. Then, the cells were stimulated with three doses of *Ba* RNA (1, 5 and 10 μg/ml) in the presence of IFN-γ. After 48 h, cells viability was checked by staining the cells with Zombie Violet™ solution. A set of heat-killed cells was used as a positive control. ****$P<0.0001$ *vs* untreated cells.

Zombie violet assay, which showed that treatment with *Ba* RNA did not significantly affect cell death in any of the cell lines (Fig 2A–2D and S1 Fig). On the other hand, other prokaryotic RNAs (from *Escherichia coli* and *Klebsiella pneumonia*) were able to down-modulate MHC-I surface expression on epithelial and endothelial cells (S2 Fig). Overall, these results indicate

that *Ba* RNA down-modulates the IFN-γ-induced surface expression of MHC-I molecules on epithelial and endothelial cells. However, this is not an exclusive phenomenon of *B. abortus* RNA as it could be extended to other prokaryotic RNAs.

## *Ba* RNA increases IL-8 and IL-6 secretion from different cell lines

Patients with chronic brucellosis show augmented levels of IFN-γ and other Th1 cytokines during active disease [32]. With respect to endothelial and epithelium cells, Ferrero *et al*. demonstrated that the infection of HMEC cells with *Ba* led to an increased production of IL-8, MCP-1, and IL-6 [18]. Also, IL-8 is augmented in *Ba*-infected Calu-6 cell supernatants but not in A549's [17]. Therefore, to determine the effect of *Ba* RNA on the secretion of pro-inflammatory cytokines, Calu-6, A-549 and HMEC cells were stimulated with three doses of *Ba* RNA (1, 5 and 10 μg/ml) in the presence of IFN-γ. After 48 h, supernatants were collected, and IL-8, IL-6 and MCP-1 secretion was quantified by sandwich ELISA. No significant change in MCP-1 secretion was observed in cells treated with *Ba* RNA (Fig 3C). Contrary to our expectations, the treatment of Calu-6, A-549 and HMEC cells with *Ba* RNA resulted in higher levels of IL-8 and IL-6 compared to untreated cells (Fig 3A and 3B). These results indicated that although *Ba* RNA affects the expression of MHC-I in the different cell lines evaluated, it increases the production of some pro-inflammatory cytokines.

## *Ba* RNA induces MHC-I retention in GA in Calu-6 and HMEC cell lines

*Ba* exhibits an immune evasion tool of down-modulating MHC-I on monocytes/macrophages in the presence of IFN-γ (when Th1 response is triggered). The overall expression of MHC-I is unchanged, but the proteins remain within the GA [11]. We have shown that *Ba* RNA reduces the surface expression of MHC-I induced by IFN-γ in other cells which are susceptible to infection with *Ba* (Fig 1). However, it is unclear whether this phenomenon is caused by *Ba* RNA-mediated MHC-I retention within the GA, as is the case in human macrophages, which is its preferred niche. To assess this, Calu-6, A-549 and HMEC cells were stimulated with 10 μg/mL *Ba* RNA in the presence of IFN-γ. At 48 h, the expression of MHC-I and the marker GM130 of GA were detected using confocal microscopy. We identified three populations: cells with bright expression of MHC-I on the cell surface (named MHC-I BRIGHT), cells with dim expression of MHC-I on the cell surface (named MHC-I DIM), and cells with no MHC-I expression on the cellular surface (named MHC-I NULL) (Fig 4A). Cells were split in two groups: MHC-I BRIGHT cells and MHC-I DIM + NULL cells. The percentage of MHC-I DIM + NULL cells was augmented in cells treated with *Ba* RNA + IFN-γ compared to those treated only with IFN-γ in all cell lines (Fig 4B). These results corroborate the decrease in MHC-I expression observed in flow cytometry experiments (Fig 1).

Afterwards, we evaluated whether this phenomenon observed in the group of cells with "(DIM + NULL) MHC-I expression" corresponded to a retention of MHC-I in the GA. In both Calu-6 and HMEC cells, *Ba* RNA induced colocalization of MHC-I and GM130 (Fig 5). However, in A-549 cells there was no evidence of colocalization (Fig 5). Taken together, these results demonstrate that the *Ba* RNA-mediated MHC-I retention in GA previously observed in human macrophages can also be reproduced in other cell lines, particularly Calu-6 and HMEC.

## *Ba* RNA-mediated MHC-I down-modulation involves the EGFR pathway in A-549 and HMEC cells

We previously demonstrated that *Ba* RNA reduces the MHC-I surface expression induced by IFN-γ on human monocytes/macrophages through a TLR8-dependent mechanism and the

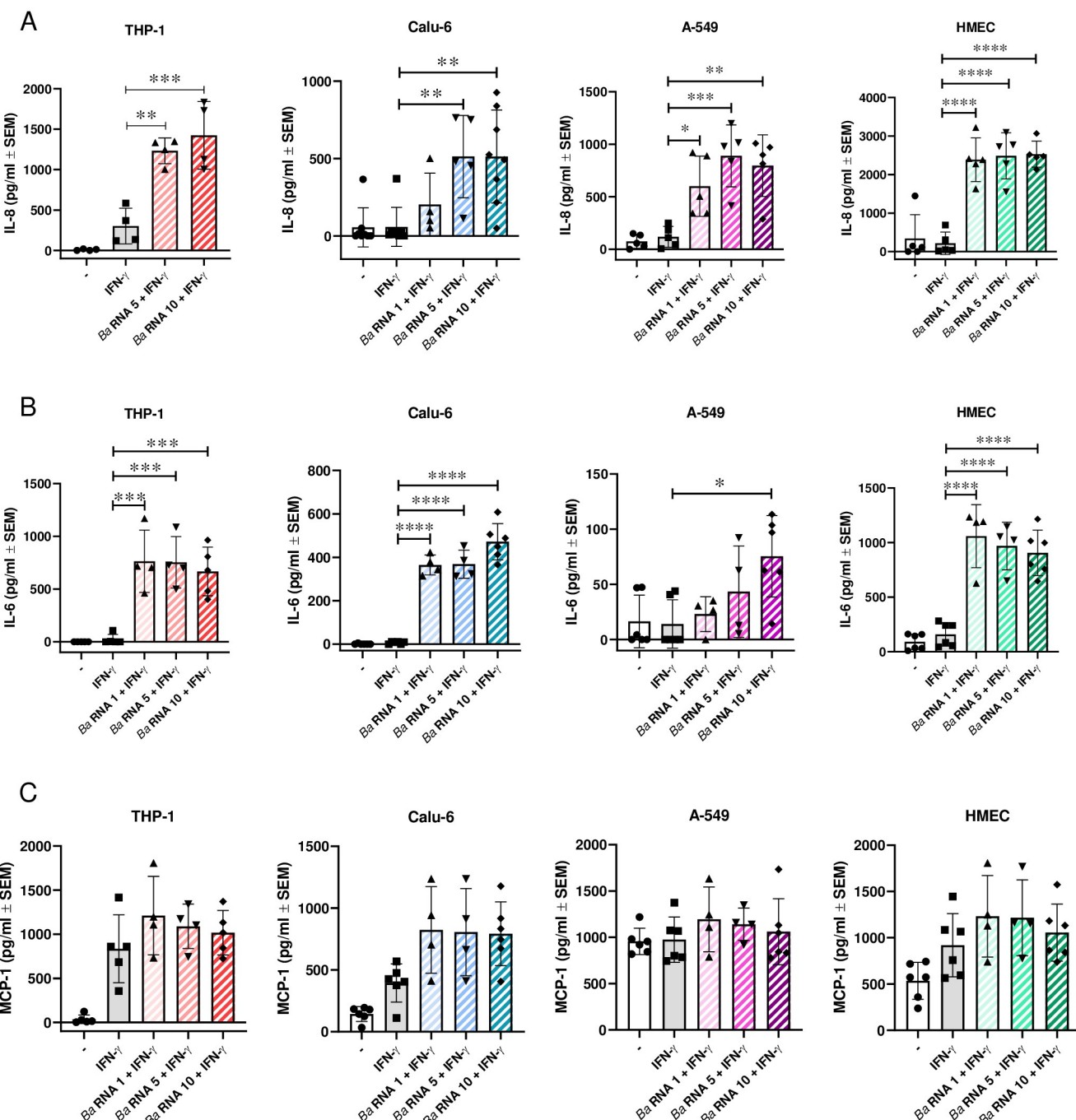

**Fig 3. *Ba* RNA increases the secretion of IL-8 and IL-6, but not MCP-1, from different cell lines.** THP-1, Calu-6, A-549 and HMEC were attached on a 48-well plate for 24 h. Then, the cells were treated with three doses of *Ba* RNA (1, 5 and 10 μg/ml) in the presence of IFN-γ. After 48 h, supernatants were analyzed for the secretion of (A) IL-8, (B) IL-6 and (C) MCP-1 by sandwich ELISA. IFN-γ-treated cells were used as a positive control. The geometric means ± SEM of at least four independent experiments are indicated in the bars. -, untreated cells. *$P < 0.05$; **$P < 0.01$; ***$P < 0.001$; ****$P < 0.0001$ *vs* IFN-γ-treated cells.

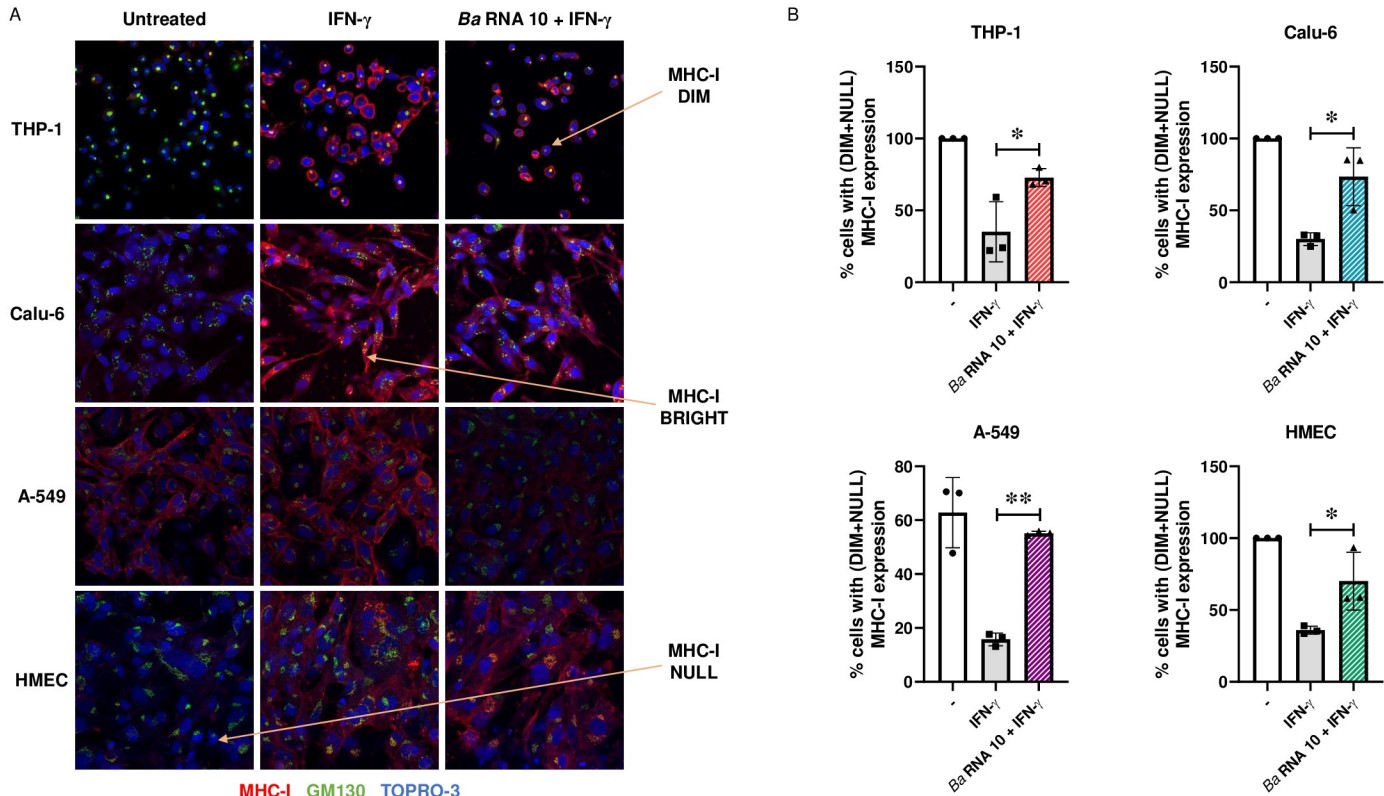

**Fig 4. *Ba* RNA increases the percentage of cells with (DIM + NULL) MHC-I expression.** (A) Confocal microscopy images of THP-1, Calu-6, A-549 or HMEC cells treated with *Ba* RNA (10 μg/ml) in the presence of IFN-γ for 48 h. MHC-I expression was detected with an anti-human MHC-I primary Ab (W6/32) followed by Alexa 546-labeled secondary Ab (red). GA was determined using a mAb specific for GM130 and Alexa 488-labeled secondary Ab (green). Nucleus was detected using TOPRO-3 (blue). Orange arrows show representative cells of the different MHC-I expression (MHC-I DIM, MHC-I BRIGHT and MHC-I NULL cells). Cells treated with IFN-γ were used as a positive control. (B) Bars represent the percentage of cells with (DIM + NULL) MHC-I expression. -, untreated cells. *P<0.05; **P<0.01 *vs* IFN-γ-treated cells. The number of cells counted per experimental group was 200.

EGFR signaling pathway [13]. To expand these results, Calu-6, A-549 and HMEC cells were stimulated with *Ba* RNA (10 μg/ml) in the presence of IFN-γ. After 48 h, the expression of TLR-8 in all cell lines was corroborated by flow cytometry (S3 Fig). Subsequently, cells were treated with *Ba* RNA (10 μg/ml) in the presence of 50 μg/ml of Cetuximab (a ligand-blocking antibody of the EGFR). Neutralization of the EGFR partially reversed the down-modulation of MHC-I surface expression mediated by *Ba* RNA in A-549 and HMEC cells (Fig 6C and 6D). These results demonstrate that the EGFR pathway is involved in the down-modulation of MHC-I mediated by *Ba* RNA in A-549 and HMEC cells.

## Discussion

Brucellosis remains being a neglected and complicated disease due to multiple reasons: it is sub-diagnosed, it could progress to a chronic disease, it occurs in multiple domestic and wild animals and as consequence, it causes socio-economic losses specially in low and middle-income countries where agricultural activities are fundamental [33]. The disease has been classified as "one of the seven neglected endemic zoonotic diseases of particular interest" by the WHO [3, 4]. Consequently, studies focused on the pathogens responsible for this disease need to be expanded. Our laboratory has been mainly working in immune evasion strategies of *Ba* on monocytes/macrophages. Here, we present the very first study describing how *Ba* down-

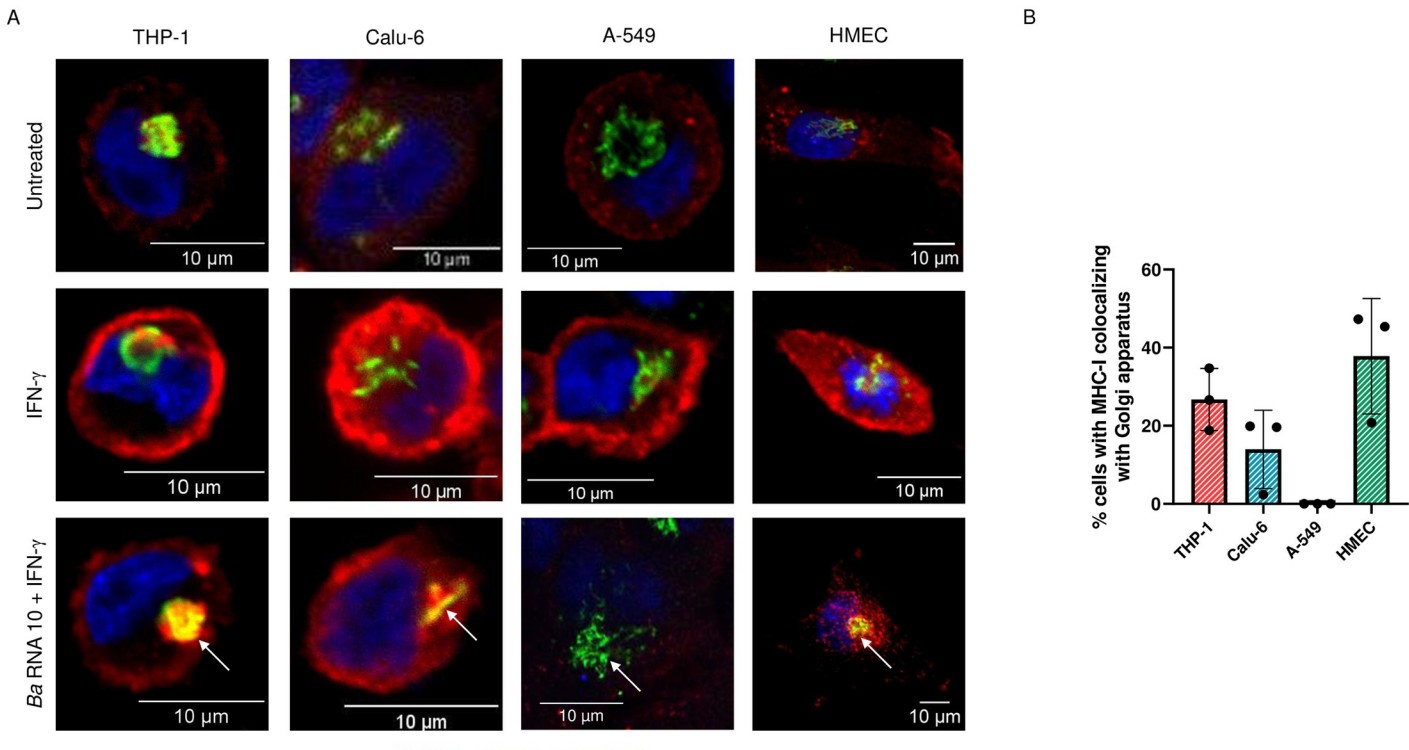

**Fig 5. *Ba* RNA reproduces the MHC-I retention in GA observed in human monocytes in Calu-6 and HMEC.** (A) Confocal microscopy images of THP-1, Calu-6, A-549 or HMEC cells treated with *Ba* RNA (10 μg/ml) in the presence of IFN-γ for 48 h. MHC-I expression was detected with an anti-human MHC-I primary Ab (W6/32) followed by Alexa 546-labeled secondary Ab (red). GA was determined using a mAb specific for GM130 and Alexa 488-labeled secondary Ab (green). Nucleus was detected using TOPRO-3 (blue). The colocalization (yellow staining) is pointed with white arrows. Cells treated with IFN-γ were used as a positive control. (B) Bars indicate the percentage of cells with "(DIM + NULL) MHC-I expression" with their MHC-I colocalizing with GA. The number of cells counted per experimental group was 200.

modulates MHC-I in cells -different from monocytes/macrophages- able to be infected with the pathogen.

Humans can get infected with *Ba* mainly through the consumption of raw dairy products, inhalation of contaminated aerosols, *e.g.* slaughterhouses and contact of injured skin with contaminated materials [1]. In line with this, respiratory and endothelial cells are crucial in the development of the infection. Ferrero *et al.* have already described that *Ba* can infect and replicate in HMEC and in human line epithelial cells [16–18]. However, it is in monocytes/macrophages where *Ba* finds its niche to persist chronically inside the host.

From the multiple resources that *Ba* employs to evade human immune response, MHC-I down-modulation is one of the most studied [11, 13, 15]. In this study, we show that *Ba* RNA-mediated MHC-I down-modulation previously described for monocytes/macrophages also occurs in human respiratory epithelial (CALU-6 and A549) and in endothelial cells (HMEC).

Reduction of MHC-I as an immune evasion strategy is a well described mechanism in virus infections [34–37]. Recently, Koutsakos *et al.* (2019) described for the first time that Influenza A and B viruses down-modulates MHC-I expression in infected cells (with A549 being one of the studied) by different mechanisms [38]. In bacterial infections, MHC-I down-regulation is not as common as for viruses. For instance, one pathogenic strain of *S. typhimurium* specifically downregulated MHC-II, but not MHC-I expression on porcine alveolar macrophages [39]. On the other hand, *Coxiella burnetii*-infected DCs show impaired DC maturation and

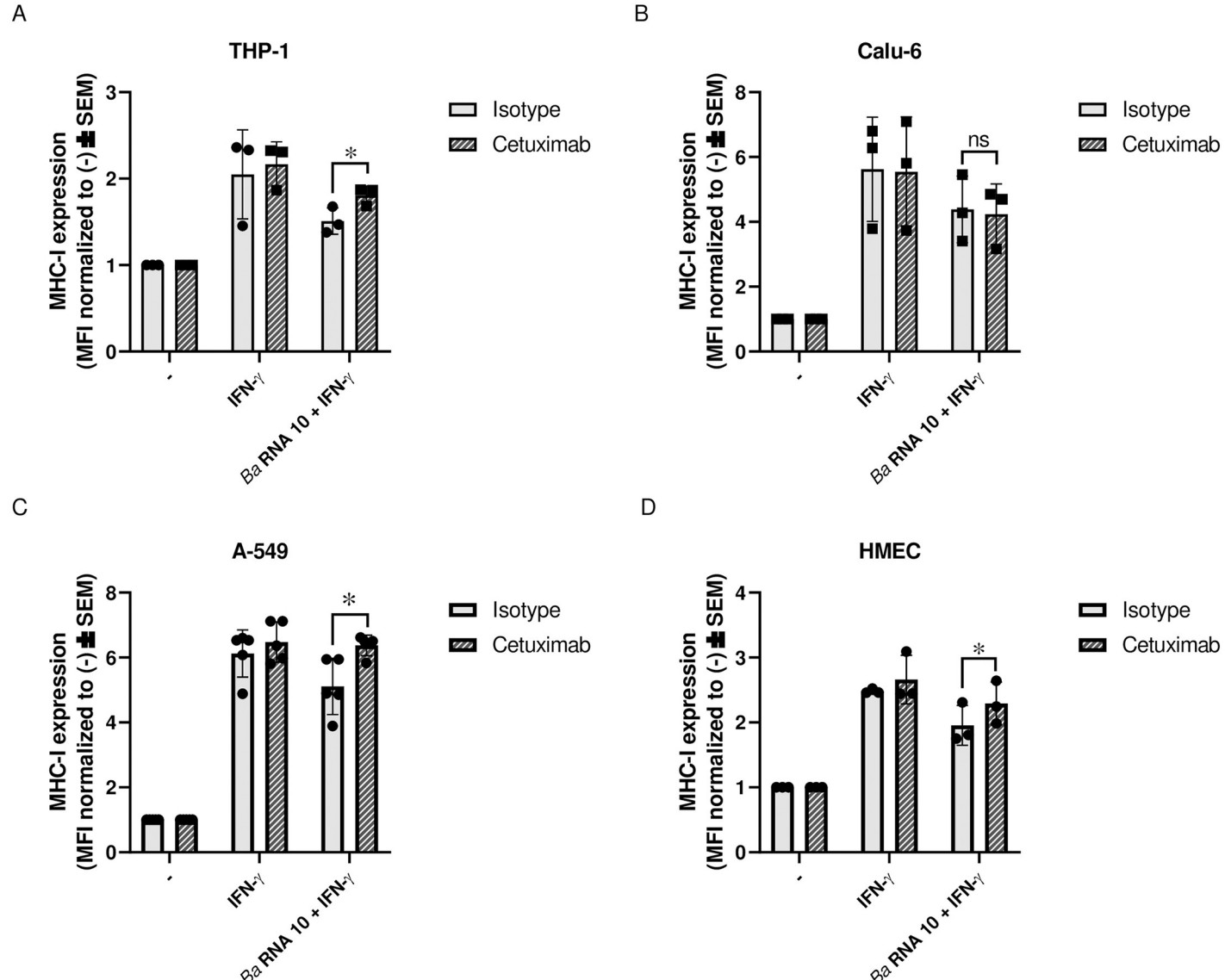

**Fig 6. Cetuximab partially reversed the MHC-I surface down-modulation mediated by *Ba* RNA in A-549 and HMEC cells.** (A) THP-1, (B) Calu-6, (C) A-549 and (D) HMEC were attached on a 48-well plate for 24 h. Then, the cells were treated with *Ba* RNA (10 μg/ml) in the presence of IFN-γ and Cetuximab (50 μg/ml) or Isotype (50 μg/ml). After 48 h, MHC-I expression was evaluated by flow cytometry. Bars represent the geometric means normalized to untreated cells ± SEM of at least three independent experiments. MFI, mean fluorescence intensity. ns, non-significant; -, untreated cells. *P<0.05 *vs* Isotype control.

MHC-I expression [40]. In our laboratory, we demonstrated that *Ba* downregulates MHC-II and MHC-I, and in both cases, *Ba* RNA is one of the PAMP that triggers this phenomenon [13, 41].

The profile of the evaluated secreted cytokines merits discussion. *Ba* RNA stimulates the IFN-γ-induced secretion of the pro-inflammatory cytokines IL-8 and IL-6 in the studied cells. Regarding the latter, we and others have shown that *Ba* infection as well as *Ba* RNA stimulates the secretion of IL-6 in the presence or absence of IFN-γ [19, 29, 41, 42]. Indeed, IL-6 is the mediator of IFN-γ-induced MHC-II down-modulation in monocytes by lipoproteins [42] and by *Ba* RNA [41]. Thus, it may not be surprising its role in this disease as a pro-inflammatory

molecule, not only in phagocytes but in endothelial and epithelial cells equally. In this regard, Mitroulis *et al*. (2022) demonstrated that in a cohort of brucellosis patients, the levels of IFN-γ, IL-1β and IL-6 diminished significantly after treatment, while there was no difference in other measured cytokines such as MCP-1 [32].

About the retention of MHC-I molecules in the GA some highlights must be commented. In previous studies from our laboratory [11, 13], we showed that *Ba* and *Ba* RNA diminished the IFN-γ-induced MHC-I expression in human monocytes by retaining these molecules in the GA. This is not due to a diminished MHC-I expression. Moreover, since we used W6/32 as the anti-HLA-ABC class I mAb, we know that detected MHC-I are properly assembled. Evidence of this phenomenon is very limited in the literature, for what this is in intriguing aspect of the mechanisms that this pathogen employs. *Ba* is not the unique pathogen capable of retaining MHC-I molecules in GA. The bovine papillomavirus affects MHC-I localization by retaining these molecules within GA in embryonic bovine cells [34]. Varicella-zoster virus impaired the traffic of MHC-I from GA to the cell membrane [43].

Regarding the receptor, TLR8 has been recently characterized as being involved in infectious diseases. For instance, the detection of Streptococcus pyogenes RNA by this receptor is critical for triggering monocyte activation in response to the infection [44]. Also, Cervantes *et al*. demonstrated that the transcription of IFN-β in response to *Borrelia burgdorferi*, is induced from within the phagosome vacuole through the TLR8-MyD88 pathway [45, 46]. We showed that human TLR8/murine TLR7 senses RNA from *Ba* promoting MHC-I down-modulation in monocytes and human/murine macrophages [13]. In this manuscript, although we are not demonstrating a direct sensing of *Ba* RNA via TLR8 in the studied cell lines, we show its presence by flow cytometry.

We have previously shown that Cetuximab, a drug that blocks the EGFR, partially reverted MHC-I down-modulation by *Ba* and its RNA in monocytes/macrophages [13, 15]. In the present study, we wondered whether this drug was able to cause the same effect in other cell lines. Interestingly, we found that Cetuximab perform the same partial reversion in A549 and HMEC cell lines. For Calu-6 is particularly remarkable the fact that cells showed retention of MHC-I molecules within the GA, but Cetuximab did not even partially revert the phenomenon. These results led us postulate that, like in monocytes/macrophages, there would be a connection between TLR8 and EGFR, at least in A549 and HMEC. To our knowledge, this is the first report describing a link between these pathways in these alternative niches of infection.

Some limitations of this study should be considered. Brucellosis patients in Argentina are sub diagnosed; a reason why obtaining a blood sample is a difficult task. Murine models are useful to study this disease; however, we have not been able to carry out trials because the unique animal BSL-3 laboratory located in Buenos Aires, Argentina, is not available due to renovations after the COVID-19 pandemic. For future research work, we would like to find a representative number of samples from patients with brucellosis to perform at least cytokine measurements and detection of MHC molecules in PBMCs, which would help us deepen our understanding of the pathology. An *in vivo* model with *Ba*-infected mice is also desired to obtain different types of cells such as lung epithelial and endothelial cells, osteoblasts, synoviocytes, hepatocytes, astrocytes and microglia, among others, capable of being infected by *Ba* to evaluate their immune profile and functionality.

Overall, our results demonstrate that *Ba* and particularly its RNA down-modulates MHC-I on its preferential niche and in other potential focuses of infection such as endothelium and respiratory epithelium. This could propose new understanding in how the pathogen may persist indefinitely pleiotropically in the host evading T CD8+ surveillance. Also, we show that Cetuximab could contribute to revert MHC-I down-modulation in other relevant cells in

brucellosis. These results expand our findings in macrophages and support the fact that Cetuximab is a promising drug to be, at least considered, for models of chronic brucellosis.

## Supporting information

**S1 Fig.** *Ba* **RNA does not induce loss of cell viability.** A-549 were attached on a 48-well plate for 24 h. Then, the cells were stimulated with *Ba* RNA (10 μg/ml) for 48 h in the presence of IFN-γ. They were then stained with Zombie Violet™ solution to check the viability of the treated cells. A set of heat-killed cells were used as a positive control. This is a representative flow cytometry histogram of the results show in Fig 2.
(TIF)

**S2 Fig. The RNA from** *Escherichia coli* (*Ec*) **and** *Klebsiella pneumoniae* (*Kp*) **also decreases the IFN-γ-induced MHC-I expression in epithelial and endothelial cells.** THP-1, Calu-6, A-549 and HMEC were attached on a 48-well plate for 24 h. Then, cells were stimulated with *Ec* or *Kp* RNA (10 μg/ml) in the presence of IFN-γ. After 48 h, flow cytometry was used to detect MHC-I expression. IFN-γ-treated cells were used as a positive control. Bars represent geometric means normalized to untreated cells ± SEM from three independent experiments. MFI, mean fluorescence intensity. -, untreated cells. $*P<0.05$; $**P<0.01$ *vs* IFN-γ-treated cells.
(TIF)

**S3 Fig. TLR-8 is expressed in different cell lines.** THP-1, Calu-6, A-549 and HMEC cells were attached on a 48-well plate for 24 h. Then TLR-8 expression was determined by flow cytometry. (A) These are representative flow cytometry histograms of results shown in panel B. (B) Bars indicate the geometric means ± SEM of four independent experiments. MFI, mean fluorescence intensity.
(TIF)

## Acknowledgments

We thank the staff of the UOCCB (Unidad Operativa Centro de Contención Biológica), ANLIS-Malbrán (Administración Nacional de Laboratorios e Institutos de Salud Dr. Carlos G. Malbrán) (Buenos Aires, Argentina) for facilitating us the use of the BSL-3 laboratory. We are very grateful with Dr. Federico Fuentes for technical support with the analysis of confocal microscopies.

## Author Contributions

**Formal analysis:** M. Ayelén Milillo, Paula Barrionuevo.

**Investigation:** Agustina Serafino, Yasmín A. Bertinat, Jorgelina Bueno, José R. Pittaluga, Federico Birnberg Weiss, M. Ayelén Milillo.

**Methodology:** Agustina Serafino, Yasmín A. Bertinat, José R. Pittaluga.

**Resources:** M. Ayelén Milillo, Paula Barrionuevo.

**Supervision:** M. Ayelén Milillo.

**Visualization:** Agustina Serafino.

**Writing – original draft:** Agustina Serafino, Paula Barrionuevo.

**Writing – review & editing:** M. Ayelén Milillo, Paula Barrionuevo.

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
