## [Decision Letter · Decision Letter 0]

25 Apr 2024

PONE-D-24-07376Beyond its preferential niche: *Brucella abortus* RNA down-modulates the IFN--induced expression of MHC-I in cells other than macrophages.PLOS ONE

Dear Dr. Milillo,

Thank you for submitting your manuscript to *PLOS ONE*. As you will see below, I received two very different reviews of your manuscript. After reading the reviewers' comments and reviewing the paper myself, I think that Reviewer 2's concerns can be readily addressed by revising the text and providing some important clarifications regarding the biological relevance of the experimental questions being addressed and how the results obtained are being interpreted. Doing this will help the general reader better appreciate the importance of the work. Please submit your revised manuscript by Jun 09 2024 11:59PM. If you will need more time than this to complete your revisions, please reply to this message or contact the journal office at plosone@plos.org. Please include the following items when submitting your revised manuscript:A rebuttal letter that responds to each point raised by the academic editor and reviewer(s). You should upload this letter as a separate file labeled 'Response to Reviewers'.A marked-up copy of your manuscript that highlights changes made to the original version. You should upload this as a separate file labeled 'Revised Manuscript with Track Changes'.An unmarked version of your revised paper without tracked changes. You should upload this as a separate file labeled 'Manuscript'.

I look forward to receiving your revised manuscript!

Sincerely,

Marty Roop

Academic Editor

*PLOS ONE*

Journal Requirements:

   "This work was financed by PB PICT-2020 SERIE A-00882 and MAM PICT-2020 SERIE A-00978 grants from the Agencia Nacional de Promoción Científica y Tecnológica (ANPCYT-Argentina)." 

Reviewers' comments:

Reviewer's Responses to Questions

**Comments to the Author**

1. Is the manuscript technically sound, and do the data support the conclusions?

Reviewer #1: Yes

Reviewer #2: No

2. Has the statistical analysis been performed appropriately and rigorously? 

Reviewer #1: Yes

Reviewer #2: Yes

3. Have the authors made all data underlying the findings in their manuscript fully available?

Reviewer #1: Yes

Reviewer #2: Yes

4. Is the manuscript presented in an intelligible fashion and written in standard English?

Reviewer #1: Yes

Reviewer #2: Yes

5. Review Comments to the Author

Reviewer #1: Brucella abortus-derived RNA (Ba RNA) is known to downregulate expression of MHCI on macrophages. Here the authors show that Ba RNA downregulates MHCI surface expression on epithelial and endothelial cells. In some cell lines, this effect was associated with retention of MHCI in the golgi. In addition, blockade of EGFR partially negated the effects of Ba RNA on MHCI surface expression. Overall, the study is generally sound and adds to the literature indicating that Ba RNA affects host responses.

1. I suggest including epithelial and endothelial cells in the title rather than “non-macrophages”.

2. Line 43-44 “… this is the first study exploring immune evasion strategies beyond monocytes….”. This statement is not entirely accurate, as Brucella is known to evade of variety of immune responses (ex. by neutrophils, dendritic cells, B cells etc).

3. Lines 145-147 “.. experiments were performed at 37C/5%CO2 and a standard medium consisting of RPMI 1640 or DMEM….”. The authors should clarify when they used media containing RPMI 1640 or DMEM”.

4. What concentration of EGFR antibody was used?

5. Why are there no error bars in Figure 2?

6. In line 269 of the Figure 3 legend the text indicates that geometric means are shown and the authors also define MFI in the legend. Is this correct or a typo resulting from carryover of text from the flow-cytometry figure legends?

7. How many cells counted were counted to generate the data in Figure 4 and Figure 5?

8. TLR8 is typically an endosomal TLR. Were the cells in Figure S2 permeabilized in order to detect TLR8 by flow cytometry?

Reviewer #2: The manuscript by Serafino and colleagues describes the phenomenon of immune modulation in cells treated with RNA from the bacterial pathogen Brucella abortus. These bacteria predominantly reside in cells such as macrophages, but the authors chose to examine other cell types, including human bronchial epithelial cells (Calu-6), human alveolar epithelial cells (A-549), and endothelial cells (HMEC). RNA isolated from B. abortus was incubated with these cells, and the authors observed MHC-I was retained in the Golgi apparatus in Calu-6 and HMEC cells, but this did not occur in A-549 cells. Additionally, treatment of the cells with RNA induced the production of specific cytokines, IL-6 and IL-8.

Overall, the conclusions drawn by the authors are not supported by the data, and moreover, the scientific premise for the work is not well described. The specific concerns are as follows:

-The rationale for performing the study is not clear. The authors seem to argue that because there are very limited respiratory manifestations of Brucella infections, this means that the epithelial cells of the respiratory track must be infected by the bacteria. In lines 349-351, the authors state that these cells are "able to be infected with the pathogen." The data do not show this, and there are no citations to support the claim that the cell lines used support Brucella infection and/or replication. As such, it is difficult to understand why the studies were undertaken, but more importantly, it is difficult to know what the data mean in terms of Brucella biology.

-The rationale for putting naked Brucella RNA onto cells and evaluating the immune response is not clear. When would cells of any type be exposed to microgram quantities of Brucella RNA?

-The RNA was isolated using an organic reagent (i.e., Trizol), but there are no controls in the experiments to rule out the possibility that organic contamination or other reagent from the isolation method is what is actually stimulating the responses that were observed. An appropriate control would likely be RNA isolated using the same method from another bacterium (e.g., E. coli). This would address if this was a general RNA phenomenon or something specific to Brucella RNA isolated using that protocol.

6. PLOS authors have the option to publish the peer review history of their article (what does this mean?). If published, this will include your full peer review and any attached files.

Reviewer #1: No

Reviewer #2: No

---

## [Author Response · Author response to Decision Letter 0]

13 Jun 2024

Reviewer #1: 

Brucella abortus-derived RNA (Ba RNA) is known to downregulate expression of MHCI on macrophages. Here the authors show that Ba RNA downregulates MHCI surface expression on epithelial and endothelial cells. In some cell lines, this effect was associated with retention of MHCI in the golgi. In addition, blockade of EGFR partially negated the effects of Ba RNA on MHCI surface expression. Overall, the study is generally sound and adds to the literature indicating that Ba RNA affects host responses.

1. I suggest including epithelial and endothelial cells in the title rather than “non-macrophages”.

As suggested by the reviewer, the title “Beyond its preferential niche: Brucella abortus RNA down-modulates the IFN-γ-induced expression of MHC-I in cells other than macrophages” has been changed to “Beyond its preferential niche: Brucella abortus RNA down-modulates the IFN-γ-induced MHC-I expression in epithelial and endothelial cells”.

2. Line 43-44 “… this is the first study exploring immune evasion strategies beyond monocytes….”. This statement is not entirely accurate, as Brucella is known to evade of variety of immune responses (ex. by neutrophils, dendritic cells, B cells etc).

We agree with the reviewer on this point. The paragraph is not entirely accurate, for this reason, it was changed to “...this is the first study exploring a central immune evasion strategy, such as the downregulation of MHC-I surface expression, beyond monocytes...” (Lines 42-44, Page 2 of the revised version of the manuscript).

3. Lines 145-147 “…experiments were performed at 37C/5%CO2 and a standard medium consisting of RPMI 1640 or DMEM…”. The authors should clarify when they used media containing RPMI 1640 or DMEM”.

As suggested by the reviewer, we clarify which medium was used for each cell line. These sentences were incorporated in the revised version of the manuscript (Lines 146-147, Page 7 of the new version of the manuscript).

4. What concentration of EGFR antibody was used?

The final concentration of EGFR antibody used was 50 microgram/milliliter (μg/ml). The same concentration was used for the corresponding isotype. This information was incorporated in the new version of the manuscript (Lines 217-218, page 10; line 339, page 15 and lines 347-348, page 15).

5. Why are there no error bars in Figure 2?

The figure has no error bars since the experiment had been performed twice. We ran another set of experiments so we could set the error bars, as shown in the new Figure 2.

6. In line 269 of the Figure 3 legend the text indicates that geometric means are shown and the authors also define MFI in the legend. Is this correct or a typo resulting from carryover of text from the flow-cytometry figure legends?

We would like to thank the reviewer for pointing out the mistake. Effectively, this is a typo resulting from the carryover of the text from the flow cytometry figure legends. The definition for MFI was eliminated from the legend of Figure 3 (Line 281, page 12).

7. How many cells counted were counted to generate the data in Figure 4 and Figure 5?

The cells counted to generate the data in Figure 4 and Figure 5 were 200. This information was incorporated in the revised version of the manuscript (Lines 311-312, page 14 and line 330, page 14). 

8. TLR8 is typically an endosomal TLR. Were the cells in Figure S2 permeabilized in order to detect TLR8 by flow cytometry?

As mentioned by the reviewer, TLR8 is an endosomal receptor. For this reason, the cells in Figure S2 (of the old version of the manuscript, S3 in the revised version) were permeabilized to detect TLR8 by flow cytometry. This clarification was incorporated in the revised version of the manuscript (Lines 179-180 and lines 185-188, page 8).

Reviewer #2: 

The manuscript by Serafino and colleagues describes the phenomenon of immune modulation in cells treated with RNA from the bacterial pathogen Brucella abortus. These bacteria predominantly reside in cells such as macrophages, but the authors chose to examine other cell types, including human bronchial epithelial cells (Calu-6), human alveolar epithelial cells (A-549), and endothelial cells (HMEC). RNA isolated from B. abortus was incubated with these cells, and the authors observed MHC-I was retained in the Golgi apparatus in Calu-6 and HMEC cells, but this did not occur in A-549 cells. Additionally, treatment of the cells with RNA induced the production of specific cytokines, IL-6 and IL-8. Overall, the conclusions drawn by the authors are not supported by the data, and moreover, the scientific premise for the work is not well described. The specific concerns are as follows:

1-The rationale for performing the study is not clear. The authors seem to argue that because there are very limited respiratory manifestations of Brucella infections, this means that the epithelial cells of the respiratory track must be infected by the bacteria. In lines 349-351, the authors state that these cells are "able to be infected with the pathogen." The data do not show this, and there are no citations to support the claim that the cell lines used support Brucella infection and/or replication. As such, it is difficult to understand why the studies were undertaken, but more importantly, it is difficult to know what the data mean in terms of Brucella biology.

We agree with the reviewer regarding the importance of clearly stating the rationale of a study to understand why the studies are carried out and what emerges from them. That is why in the introduction of our manuscript we particularly emphasized the ability of the bacterium to impact on a wide variety of organs and tissues ("This condition is multisystemic affecting different tissues and organs” Page 3, lines 63-64). Furthermore, we explained that although the monocytes/macrophages are the preferential intracellular niche for the bacterium, Ba can infect other cells: 

“Ba can also infect and replicate in different cell types such as lung epithelial and endothelial cells, osteoblasts, synoviocytes, hepatocytes, astrocytes, and 

microglia, among others [16-24] (Page 4, lines 92-94)”. 

Among the mentioned citations, precisely the citations 16 and 17 demonstrate that Ba is capable of infecting human lung epithelial cells. More specifically, human bronchial epithelium (Calu-6) and human alveolar epithelium (A-549) cell lines. Citations are mentioned here:

16. Ferrero MC, Fossati CA, Baldi PC. Smooth Brucella strains invade and replicate human lung epithelial cells without inducing cell death. Microbes Infect. 2009;11(4):476–83. 

17. Ferrero MC, Fossati CA, Baldi PC. Direct and monocyte-induced innate immune response of human lung epithelial cells to Brucella abortus infection. Microbes Infect. 2010;12(10):736–47. 

For its part, the citation number 18 demonstrates that Ba is capable of infecting human endothelial cells. More specifically human endothelial microvasculature (HMEC) and human umbilical vein endothelial cells (HUVEC) cell lines. Citation is mentioned here:

18. Ferrero MC, Bregante J, Delpino MV, Barrionuevo P, Fossati CA, Giambartolomei GH, Baldi PC. Proinflammatory response of human endothelial cells to Brucella infection. Microbes Infect. 2011 ;13(10):852–61. 

Added to this, in our rationale to explain the study, what we wanted to denote in the introduction is the importance of the epithelium and the endothelium in the pathogenesis of the infection. Since the inhalation of contaminated aerosols is a central route in Ba infection, the lung epithelium plays a critical role in brucellosis (this is explained on Page 5, lines 95-105). On the other hand, most of the mortality cases in brucellosis are associated with endocarditis, denoting the great importance of the endothelium in the pathology (this is explained in Page 5, lines 106-113).

However, despite the crucial relevance of lung epithelial and endothelial cells in the context of brucellosis pathogenesis, up to now, there have not been studies of Ba immune evasion strategies on those cells. Therefore, in this study we proposed for the first time to investigate the long-term persistence of the disease in various organs of the infected host.

Overall, the results of our study demonstrate that Ba and particularly its RNA down-modulates MHC-I on its preferential niche and in other potential focuses of infection such as endothelium and respiratory epithelium. This may propose a new understanding on how the pathogen persists indefinitely pleiotropically in the host evading T CD8+ surveillance.

2-The rationale for putting naked Brucella RNA onto cells and evaluating the immune response is not clear. When would cells of any type be exposed to microgram quantities of Brucella RNA?

In relation to the reviewer's question, we will here try to explain the rationale of why introducing naked B abortus RNA into the studied cells.

We have previously reported that infection of human monocytes/macrophages with B. abortus inhibits the IFN-γ-induced MHC-I cell surface expression. This phenomenon is dependent on bacterial viability as was demonstrated by the incapacity of heat-killed bacteria to inhibit the expression of such molecules. This led us to postulate that the B. abortus component(s) involved in MHC-I inhibition corresponded to vita-PAMPs, such as bacterial RNA, as they are only expressed when bacteria are metabolically active (Milillo et al., 2017; Mourao-Sa et al., 2013; Sander et al., 2011; Ugolini & Sander, 2019). However, it is interesting to explain when and in which quantities bacterial RNA may encounter cells.

Firstly, we will cite the discussion of our own manuscript (Milillo et al., 2017), to explain how, during Ba infection, the human monocyte/macrophages can sense bacterial RNA:

“It has been described that viral and bacterial RNA are sensed by pattern recognition receptors (PRRs), among which the TLRs family has gained more attention (Mogensen, 2009). TLR3, TLR7 and TLR8 are the ones preferentially expressed in intracellular vesicles of the endoplasmic reticulum (ER), endosomes, and lysosomes (Mifsud et al., 2014). With respect to the intracellular cycle of the bacterium, Ba can enter, survive and replicate within vacuolar phagocytic compartments of macrophages (Gorvel & Moreno, 2002). Once inside the macrophages, Brucella dwells in an acidified compartment that fuses with components of the early and late endosomal/lysosomes pathway (Starr et al., 2008). There, most of the ingested bacteria are rapidly killed. However, the establishment of a persistent infection depends on the ability of the bacterium to form a Brucella containing vacuole (BCV), which traffics from the endocytic compartment to the endoplasmic reticulum (ER) (Gorvel & Moreno, 2002; Roop 2nd et al., 2004). Once inside the replicative BCV, bacteria are resistant to further attack and begin to multiply dramatically (Roop 2nd et al., 2004). Starr et al demonstrated that Brucella replication in the ER is followed by BCV conversion into multi-membrane LAMP-1-positive vacuoles with autophagic features (aBCV). Furthermore, aBCVs were required to complete the intracellular Brucella lifecycle and for cell-to-cell spreading (Starr et al., 2012). In this context, it is possible that while Ba traffics through early and late endosomes/lysosomes the bacterial RNA released during phagocytosis activate endolysosomal TLRs. On the other hand, Ba mutant strains on virulence factors are also capable of infecting human monocytes/macrophages and transiting by early and late endosomes/lysosomes, but, unlike wild-type Ba, they are unable to replicate in BCVs and thus persist into the cell host. However, our results demonstrated that these strains are equally able to down-regulate MHC-I than wildtype Ba. Therefore, the RNA of these bacteria could also gain access to TLR3, TLR7 or TLR8 in their transit through endosomes and lysosomes, although they do not persist in macrophages. In accordance with this, it was reported that human TLR8 is activated upon recognition of Borrelia burgdorferi RNA in the phagosome of human monocytes (Cervantes et al., 2013). In our in vitro experiments of stimulation with purified RNA, either in the presence or the absence of transfection, the entry of RNA by endocytosis gaining access to the endosomal TLRs can perfectly mimic what happens in an infectious context.”

During a chronic infection, as in Ba’s, bacteria may reach the blood flow, causing a bacteriemia. (Young & Corbel, 1989). The periods of bacteriemia take place in patients with acute and chronic brucellosis. (Pappas & Papadimitriou, 2007). Therefore, it may be possible that these bacteria release their RNA to the extracellular media; in fact, prokaryotic RNA has been observed in media cultures where bacteria are alive, suggesting an active mechanism of RNA release (Ghosal et al., 2015; Lee & Tan, 2014; Obregón-Henao et al., 2012). Recently, prokaryotic RNA has been identified in mammals’ blood (Freedman et al., 2016; Wang et al., 2012).

Moreover, our work group recently demonstrated that Escherichia coli RNA and its degradation products can activate neutrophils (Rodriguez-Rodrigues et al., 2017)and endothelial cells (Castillo et al., 2019).

Regarding the doses of RNA used, we used 1-10 µg/ml. This dose is the one that it is usually employed in papers that work with RNA since it is compatible with what would be present in an infectious focus (Mourao-Sa et al., 2013; Sander et al., 2011; Ugolini & Sander, 2019). Even so, in our previous studies we observed effects with RNA degradations products of concentrations of 5 and even 1 µg/ml (Castillo et al., 2019; Rodriguez-Rodrigues et al., 2017). We must consider that for infectious experiments with Ba, the multiplicity of infection (MOI) is higher than for other intracellular bacteria, such as Mycobacterium tuberculosis (Kviatcovsky et al., 2017). Another example is Borrelia spp, for Borrelia experiments, the authors use 0.1-1 μg of Bb RNA (an amount equivalent to an MOI 10:1 of live organism) (Cervantes et al., 2013). 

In our case, experiments designed by Ferrero et al. (Ferrero et al., 2009) infections of Calu-6 and A-549 were performed at multiplicities of infection (MOI) of 200 bacteria/cell. In the case of HMEC, Brucella infections were performed at a multiplicity of infection (MOI) of 100 and 1000 (Ferrero et al., 2011). So, it is logical that the doses of Ba RNA that we used to perform our experiments are in the range of 1-10 μg/ml.

-The RNA was isolated using an organic reagent (i.e., Trizol), but there are no controls in the experiments to rule out the possibility that organic contamination or other reagent from the isolation method is what is actually stimulating the responses that were observed. An appropriate control would likely be RNA isolated using the same method from another bacterium (e.g., E. coli). This would address if this was a general RNA phenomenon or something specific to Brucella RNA isolated using that protocol.

We agreed with the reviewer that during the purification of the RNA certain phenol traces might remain in the preparation. However, controls to rule out the possibility of organic contamination were not incorporated in this study since they had already been shown in our prior publication about MHC-I modulation on monocytes/macrophages by B. abortus RNA (Milillo et al., 2017). For this, we performed a mock RNA extraction, (i. e. in the absence of bacteria) and used it as a control. This treatment was not able to down-modulate MHC-I (Fig 3A. Milillo et al., 2017). Moreover, RNA purified with Quick-RNA MiniPrep kit (a method other than TRIzol) was equally able to inhibit MHC-I expression on monocytes/macrophages (S2 Fig. Milillo et al., 2017).

In addition to phenol contamination, RNA preparation could contain traces of DNA and proteins. Despite using only RNA preparation with a ratio of absorbance 260/280 > 2.0 and a ratio of absorbance 260/230 > 1.8 (as measured using a DeNovix DS-11 Spectrophotometer and indicated in Materials and Methods), we performed rigorous controls to discard the effect of these potential contaminants. These controls were also not incorporated in this study since they had already been shown in our prior publication about MHC-I modulation on monocytes/m

---

## [Decision Letter · Decision Letter 1]

18 Jun 2024

Beyond its preferential niche:* Brucella abortus *RNA down-modulates the IFN-γ-induced MHC-I expression in epithelial and endothelial cells

PONE-D-24-07376R1

Dear Dr. Milillo,

Thank you for your conscientious attention to the reviewers' constructive comments in preparing your revision! I'm pleased to inform you that your manuscript has been judged scientifically suitable for publication and will be formally accepted for publication once it meets all outstanding technical requirements.

Sincerely,

Marty Roop

Academic Editor

*PLOS ONE*

Additional Editor Comments (optional):

Reviewers' comments:

Reviewer's Responses to Questions

**Comments to the Author**

1. If the authors have adequately addressed your comments raised in a previous round of review and you feel that this manuscript is now acceptable for publication, you may indicate that here to bypass the “Comments to the Author” section, enter your conflict of interest statement in the “Confidential to Editor” section, and submit your "Accept" recommendation.

Reviewer #1: All comments have been addressed

Reviewer #2: All comments have been addressed

2. Is the manuscript technically sound, and do the data support the conclusions?

Reviewer #1: Yes

Reviewer #2: Yes

3. Has the statistical analysis been performed appropriately and rigorously? 

Reviewer #1: Yes

Reviewer #2: Yes

4. Have the authors made all data underlying the findings in their manuscript fully available?

Reviewer #1: Yes

Reviewer #2: Yes

5. Is the manuscript presented in an intelligible fashion and written in standard English?

Reviewer #1: Yes

Reviewer #2: Yes

6. Review Comments to the Author

Reviewer #1: My concerns have been addressed, however there appears to be an errors in the Figure uploads.

Figure 2 appears to have been uploaded twice.

Two versions of Figure S2 are included in the manuscript, however one of these Figures appears to be Figure S3. The Figure S3 that is included appears to be a duplicate of Figure S2.

Reviewer #2: The authors have addressed the concerns that were raised during the initial review of the manuscript, and there are no additional concerns.

7. PLOS authors have the option to publish the peer review history of their article (what does this mean?). If published, this will include your full peer review and any attached files.

Reviewer #1: No

Reviewer #2: No

---

## [Editor Report · Acceptance letter]

29 Jun 2024

PONE-D-24-07376R1 

PLOS ONE

Dear Dr. Milillo, 

I'm pleased to inform you that your manuscript has been deemed suitable for publication in PLOS ONE. Congratulations! Your manuscript is now being handed over to our production team.

Kind regards, 

on behalf of

Dr. Roy Martin Roop II 

Academic Editor

PLOS ONE